# Precision Livestock Farming Applied to Swine Farms—A Systematic Literature Review

**DOI:** 10.3390/ani15142138

**Published:** 2025-07-19

**Authors:** Aldie Trabachini, Michele da Rocha Moreira, Érik dos Santos Harada, Magno do Nascimento Amorim, Késia Oliveira da Silva-Miranda

**Affiliations:** 1Faculdade de Tecnologia de São Roque, São Roque 18130-070, SP, Brazil; 2Escola Superior de Agricultura “Luiz de Queiroz”, Universidade de São Paulo, Piracicaba 13418-900, SP, Brazil

**Keywords:** animal welfare, swine production, precision animal welfare technologies

## Abstract

This research delved into the application of precision technologies in pig farming through the analysis of 75 scientific articles. Using an innovative methodology that included the formation of four thematic groups and a SWOT analysis, we identified the main trends and challenges in the field. The results eveal the potential of precision livestock farming to optimize production, improve animal welfare, and reduce environmental impact. However, the lack of standardized metrics and the need for greater investment in research are obstacles to be overcome. The research highlights the importance of integrating technologies such as sensors, cameras, and artificial intelligence to monitor individual animals and ensure adequate welfare conditions. This approach allows for more accurate and efficient decision-making, contributing to the sustainability of the pig sector.

## 1. Introduction

Modern swine production faces increasing pressure to meet high standards of animal welfare, production efficiency, and environmental sustainability [1,2]. However, the literature reveals a series of significant gaps that limit progress in this area. First, the lack of standardized animal welfare metrics makes consistent assessment and monitoring of animal conditions challenging, reducing the scientific validation of technologies applied in swine farming [3,4,5]. The acceptance of Precision Livestock Farming (PLF) technology by producers remains an obstacle, often due to high initial costs and the perceived complexity of these technologies [5].

These challenges indicate an urgent need for research to address these limitations, promoting both the development of validated animal welfare indicators and strategies to facilitate the adoption of PLF. This study aims to respond to these demands by offering a comprehensive analysis of the advancements and gaps in the use of precision technologies in swine production and suggesting pathways to overcome the identified barriers [5].

Animal welfare is a growing concern in swine production, with ethical, economic, and regulatory implications. PLF technologies, such as sensors, cameras, and machine learning algorithms, propose the possibility of continuous and real-time monitoring of animal behavior and health individually, providing early warnings about compromised welfare situations [3,4,5,6,7,8]. These technologies significantly improve the detection of health and behavioral problems, enabling faster and more effective interventions [1,9,10,11].

PLF has a positive economic impact on pig farms by improving zootechnical performance and reducing production costs [12,13]. However, the adoption of these technologies faces significant challenges, including the need for the development and dissemination of new methodologies and acceptance by farmers [14,15]. Successful implementation of these practices can lead to more efficient and sustainable production, meeting the growing demand for food and consumer expectations for more responsible farming practices [16,17].

The use of precision feeding systems, which allow the daily adjustment of pigs’ diets according to their individual needs, results in a reduction in feed costs by over 8% and a decrease in nutrient excretion by almost 40% [10,18,19,20]. The adoption of PLF also has the advantage of significantly reducing greenhouse gas (GHG) emissions and pollution from nitrates and antibiotics, contributing to the environmental sustainability of farms [13,14,21].

The main challenge of PLF is transforming the large amount of data collected into useful and accessible information for farmers [18]. The lack of external validation for many commercially available PLF technologies limits both their adoption and effectiveness [14,22]. The concept of Industry 4.0 involves, among other aspects, horizontal (off-farm) and vertical (on-farm) integration of information, and the evolution of PLF will allow the adoption of these concepts, expanding horizons for the management of production processes and improving the assessment of animal welfare and health [23,24].

A systematic review is a valuable tool for understanding and improving pig welfare in production. Through it, effective practices can be identified, emerging technologies evaluated, and producers’ perceptions understood. Understanding academic approaches related to best practices, the use of precision technologies, and adherence to guidelines is essential for understanding pig welfare at all stages of production [25,26,27,28].

This article is structured with an introduction that synthesizes the literature, revie-wing the definitions that will be addressed throughout this research. Then, the methodology and research parameters of the systematic review are presented. The results are quantitatively presented, providing information that will allow discussion and the construction of a SWOT analysis, highlighting the strengths, weaknesses, opportunities, and threats of precision livestock farming and animal welfare in pig farming. Finally, the conclusion explores opportunities and new directions for PLF.

The objective of the systematic review on the proposed topic is to identify research gaps through the analysis of the existing literature, determining areas of knowledge that are still little explored or with few studies. These gaps help to understand which aspects of precision livestock farming require further investigation. In addition, the review aims to evaluate and compare different approaches, technologies, and strategies used in precision livestock farming, seeking to identify relevant patterns, trends, and divergences. Thus, the development of this work is justified, which aims to contribute to the improvement of knowledge and allow an approach that meets the contemporary challenges of the academic and productive sectors.

## 2. Materials and Methods

A Systematic Literature Review (SLR) is an essential methodology for synthesizing existing knowledge in a research field in an organized and replicable manner. This technique is widely used to ensure that all relevant evidence is considered, minimizing information loss and providing a comprehensive view of a specific topic [29,30,31].

Mixed-methods systematic reviews allow for an approach that combines quantitative and qualitative data, providing a more complete basis for the analysis of the subject matter [32]. The planning and development of a review protocol include formulating research questions, systematic search strategies, quality assessment, data extraction, synthesis, and interpretation of information [29,33,34]. The systematic integration of search results and the critical assessment of the extent, nature, and quality of evidence concerning a specific research question are essential for building scientific knowledge [31,33].

Boolean operators AND, OR, and NOT were used in search systems to combine or exclude terms, optimizing online searches. The AND operator filters content that contains all the keywords mentioned in the search, the OR operator retrieves content with any of the inserted keywords, and the NOT operator excludes specific words from the search, making it more specific [35,36,37,38]. These operators help refine results and find relevant scientific literature. Additionally, the asterisk (*) is used in search expressions, especially in search engines and programming languages. In search engines or databases, the asterisk is used to represent any sequence of characters [39,40].

The search terms included “pig”, “swine farming”, “precision livestock farming”, “precision zootechnics”, “animal welfare”, “management systems for pig farms”, “indicator management”, “fuzzy logic”. The search terms and operators were applied to the title, abstract, and keywords as follows: (pig OR swine farming OR precision livestock farming OR precision zootechnics OR animal welfare OR management systems for pig farms OR indicator management OR fuzzy logic*) AND (validation* OR evaluation* OR test* OR meta-analysis* OR systematic review* OR RCT OR observational study).

The search terms were selected to cover all relevant aspects of precision livestock farming. Terms such as “pig” and “swine farming” were chosen to ensure the inclusion of studies on swine farming. “Precision livestock farming” and “precision zootechnics” were used to capture specific research on precision technologies in livestock farming. “Animal welfare” and “management systems for pig farms” were included to address aspects of animal welfare and management systems, while “fuzzy logic” was selected to include studies that use this specific technique.

Searches were conducted in the following academic databases: Journals Portal CAPES, PubMed, IEEE Xplore, Web of Science, Scopus, JSTOR, and DOAJ (Directory of Open Access Journals). These databases were chosen based on their comprehensive coverage of scientific literature, relevance to the fields of livestock farming and precision agriculture, and accessibility to high-quality, peer-reviewed research. The inclusion criteria focused on studies that provided empirical data, systematic reviews, or meta-analyses relevant to the search terms, while exclusion criteria were applied to non-peer-reviewed articles, opinion pieces, and studies not directly related to the specified search terms.

The inclusion and exclusion criteria were established to ensure the relevance and quality of the selected studies. Articles published between 2019 and 2024 in English were included to guarantee the currency of the information. Studies related to nutrition and drug use, as well as those not focused on swine farming, were excluded to maintain a clear focus on precision technologies applied to swine farming.

The sixty most relevant articles, according to the criteria of the consulted database, were verified and analyzed to meet the research purpose. To define the thematic groups, we based the approach on the Five Domains Model (FDM), a systematic method used to assess animal welfare [41,42]. This model includes nutrition, environment, health, behavior, and mental/affective state. However, to meet the inclusion and exclusion criteria of the literature review, the model was adapted to four domains, broadly integrating the research-related keywords while meeting the details and specificities [43]. The defined thematic groups are presented in Table 1.

All information was meticulously recorded, allowing for the detailed identification of the articles, their authors, DOI, thematic group, study focus, objectives, employed technologies, applied methodology, population, sample size, and obtained results. Additionally, a PRISMA (Preferred Reporting Items for Systematic Reviews and Meta-Analyses) diagram was constructed, a widely recognized and utilized tool to enhance the transparency and quality of systematic reviews and meta-analyses [44,45].

A statistical approach was employed to synthesize and analyze qualitative and quantitative data from various studies. This approach is essential for conducting systematic literature reviews, allowing for the understanding of the various applications of Precision Livestock Farming (PLF) technologies [33].

To identify and synthesize recurring themes and patterns in the reviewed studies, qualitative coding methods were applied. These themes and patterns were organized into categories and subcategories. Subsequently, meta-analysis was used to quantitatively combine the results of different studies, providing an overview of the effects of PLF technologies.

Finally, a SWOT analysis was conducted to assess the strengths, weaknesses, opportunities, and threats of the studied approaches [46,47,48,49]. The integration of these methods allowed for a more robust and comprehensive analysis, combining the depth of qualitative analysis with the quantitative precision of meta-analysis. This mixed approach enabled a more complete understanding of the various applications of PLF technologies, identifying both qualitative trends and significant quantitative effects.

## 3. Results

Articles from academic databases were analyzed for the period between 1 January 2019 and 31 July 2024. The databases provided a total of 1216 articles, from which 300 were selected for initial analysis. Of these, 98 articles met the inclusion criteria and were classified into the defined thematic groups in the first screening. After a second screening, 22 articles were excluded, resulting in 75 articles that were analyzed in the systematic review. The PRISMA flow diagram, presented in Figure 1, details the bibliographic research aspects carried out for this systematic review.

Of the researched articles, 37% fit into thematic group A (animal identification and monitoring), 24% fit into thematic group B (human–animal relationships in swine farming), 28% fit into thematic group C (Animal welfare), and 11% fit into thematic group D (productive and economic management), as shown in Figure 2.

The number of publications per year was as follows: in 2019, 15 articles were published; in 2020, 11 articles; in 2021, 20 articles; in 2022, 10 articles; in 2023, 17 articles; and in 2024, 2 articles were reviewed (Figure 3).

An analysis by thematic group was conducted to identify and synthesize the key focuses, technologies, and patterns in the reviewed studies. Qualitative methods were applied to organize themes into categories that emerged from the data, allowing for a comprehensive view of current research directions in PLF.

Table 2 summarizes the main areas of focus in studies in this field. Data were categorized into seven distinct areas: technologies for PLF, application of survey forms, survey of producer or consumer perspectives on PLF, monitoring systems, management, behavioral aspects, information systems, and Artificial Intelligence (AI). Specifically, the “Survey Forms” category includes studies that used survey questionnaires to collect insights from producers and citizens, aiming to understand their perspectives and awareness regarding animal welfare and precision livestock practices. Specifically, the category “Application of Research Forms” includes studies based on questionnaire responses addressing various productive sectors.

The analysis of articles on “Precision Livestock Farming Technologies” reveals that this is the most widely explored area, with a total of 25 studies, indicating a strong interest in applying innovative technologies to optimize livestock production efficiency. Among these studies, 12% used technologies for individualized animal analysis, 47% for collective herd analysis, 11% adopted mixed approaches (both individual and collective), and in 32% of cases, the type of approach is not specified. These findings suggest that, while there is significant interest in precision technologies, there remains a methodological diversity in approaches, reflecting a lack of standardization.

In 15 studies, the “Application of Survey Forms” was used to collect information from producers and consumers to understand their knowledge about Precision Livestock Farming (PLF). In 14 studies, the “Producer or Consumer Perspective” was examined, highlighting a growing concern in academic research with understanding social perceptions and the needs of producers in the production chain. “Monitoring and Management” was the focus of 10 studies, underscoring its importance within the context of precision livestock farming.

In the “Behavioral” area, 7 studies were conducted. Although this field is relevant to PLF, it still lacks emerging technologies for support. The area of “Information Systems and Artificial Intelligence (AI)” was represented by only 3 studies, suggesting opportunities for further research, especially in the development and application of information systems and AI in PLF.

Table 3 provides an analysis of the technologies addressed in studies on precision livestock farming. The reviewed studies employed a range of technologies for the development of PLF applications, including image analysis, neural networks and AI, algorithms, RFID (Radio Frequency Identification), IoT (Internet of Things), sensors, vocalization, as well as management and monitoring tools.

“Image Analysis” also stands out with 10 studies, highlighting significant interest in using imaging technologies to monitor and evaluate various aspects of livestock farming. The use of cameras as sensors is present in 12% of the image analysis studies. Neural networks and AI technologies are covered in 5 studies, indicating a growing interest from 2020 to 2023 in applying artificial intelligence to improve decision-making and efficiency in livestock management. The vocalization of pigs is examined in 3 studies, suggesting emerging interest in using animal sounds as indicators of welfare and behavior.

“Algorithms” were applied in 5 studies, while “RFID, IoT, and Sensors” technologies were used in another 5 studies, focusing on the individualized identification and monitoring of animals. These technologies are notable for their ability to interact and exchange data within computerized systems, facilitating real-time tracking and analysis. “Management and Monitoring” were addressed in only 3 studies, indicating an opportunity for further research on advanced management and monitoring strategies within the context of precision livestock farming.

Table 4 presents an analysis of the methodologies addressed in studies related to PLF. The data were categorized into four distinct methodologies: latest advances, model development and comparison, controlled precision observational study, and data survey and analysis.

The “Latest Advances” methodology, in its various forms, is the most utilized, appearing in a total of 37 studies. This prevalence reflects strong trends in the field of Precision Livestock Farming (PLF), underscoring its importance for identifying research gaps and guiding future studies. Following this, the “Development and Comparison of Models” methodology is featured in 26 studies, demonstrating a substantial interest in developing and evaluating diverse models to enhance precision and efficiency in livestock management.

The methodologies “Controlled Precision Observational Study” and “Research and Data Analysis” each appear in 6 studies, highlighting the importance of conducting controlled observations and detailed analyses to yield precise and reliable data.

As for the populations studied, pigs at various rearing stages (gestation, growth, and finishing) represent 31% of the research focus. Sixteen percent of the studies involve article reviews, 13% focus on producers and members of civil society, and 7% examine farm structures and operational procedures.

Table 5 provides an analysis of the types of outcomes achieved in PLF studies. The results are categorized into five distinct areas: indicators of animal welfare in pigs, equipment and systems qualification, social acceptance of PLF, pig behavior, and PLF system reliability.

The data analysis indicates that “Indicators of Animal Welfare in Pigs” are the most frequently studied outcome, with a total of 30 studies. This reflects a strong focus and concern for animal welfare within PL, emphasizing the importance of ensuring appropriate conditions for pigs. “Equipment and Systems Qualification” appears in 20 studies, suggesting significant interest in assessing and validating the technologies and systems employed in PLF to guarantee their effectiveness and efficiency.

The “Social Acceptance of PLF” is examined in 11 studies, highlighting a growing interest in understanding societal perceptions and acceptance of precision technologies in livestock farming—a critical factor for the successful implementation of these innovations. “Pig Behaviors” are explored in 10 studies, underscoring the importance of monitoring and understanding animal behavior to enhance management practices and welfare standards. Lastly, “PLF Systems Reliability” is the least studied outcome, with only 4 studies, indicating an opportunity for further research focused on ensuring the reliability and robustness of PLF systems.

## 4. Discussion

Discussion on PLF in swine production involves a complex analysis of the integration between advanced technologies, conscious management, and animal welfare. PLF technologies are being adopted not only to increase production efficiency but also to ensure more ethical and sustainable practices in animal management [3,13,17]. Sustainability seeks a balance between animal health and well-being, environmental preservation, and economic and ethical aspects, aiming for the long-term maintenance of animal production systems [3,10,13,17]. Automated monitoring, individual tracking, and the use of IoT systems emerge as essential tools for creating a more suitable environment for pigs, benefiting both the comfort and health of the animals, as well as the overall productivity of the system.

These approaches indicate a growing movement in the sector toward the adoption of animal welfare certifications and practices aligned with consumer and societal expectations, which demand greater transparency and environmental responsibility. In this con-text, the relationship between humans and animals, environmental conditions, economic management, and new technologies become central pillars for developing a more modern and conscious swine industry.

The following discussion explores each proposed thematic group, analyzing the implications of adopting PLF technologies in swine production, focusing on promoting animal welfare, economic efficiency, and sector sustainability. Sustainability, in this context, seeks a balance between animal health and well-being, environmental preservation, and economic and ethical aspects, aiming for the long-term maintenance of animal production systems. The adoption of automated monitoring, individual tracking, and the use of IoT systems emerge as essential tools for creating a more suitable environment for pigs, benefiting both the comfort and health of the animals as well as the overall productivity of the system.

### 4.1. Thematic Group A

Thematic group A, which analyzes “Animal Identification and Monitoring” within PLF, reflects the growing adoption of advanced technologies to improve efficiency, animal welfare, and process management, particularly in swine production.

Most studies focus on using advanced technologies, such as RGB cameras, IoT, and neural networks, for monitoring and identifying pigs, aiming to automate data collection and increase precision in identifying behaviors and health conditions of animals [8,50,51,52,53,54,55]. This is evident in studies exploring behavioral monitoring using cameras, as well as the use of algorithms for tracking and counting animals in different environments [52,56,57].

Several articles review or propose the application of Artificial Intelligence (AI) to solve critical problems in pig farming, such as detecting abnormal behaviors and monitoring animal welfare [26,57,58,59,60]. This trend shows the shift towards more autonomous and intelligent systems capable of analyzing large volumes of data in real time.

The concern for animal welfare is a constant in the reviewed studies, highlighting how PLF can contribute to more ethical and sustainable practices. Studies investigating the role of PLF technologies in monitoring welfare, as well as studies on technologies focused on assessing animal welfare, underline this trend [10,50,51,54,55,57,61]. There is significant emphasis in studies, suggesting a continuous effort to consolidate existing knowledge and identify gaps for future research [6,9,10,16,17,26,60,62,63,64,65,66].

Although PLF technologies are promising, significant challenges exist in validating positive welfare indicators and adopting these technologies by producers [54,57,67]. The lack of robust evidence on the effectiveness of welfare indicators, such as positive affective states, limits the ability of these technologies to promote effective animal welfare [64]. Additionally, concerns are raised that focusing on health and productivity may lead to a restrictive definition of welfare, ignoring animals’ mental and emotional aspects [52,53,65].

Another relevant aspect is the monitoring of the perception and acceptance of these technologies by farmers and society in general. Social pressure and policies significantly influence the adoption of new animal monitoring technologies to quantify animal welfare within the production process and to adopt sustainability criteria [54,67].

### 4.2. Thematic Group B

Thematic group B analyzes the “Human–Animal Relationship in Swine Production,” especially regarding the understanding and use of animal welfare concepts, both by producers and society in general [68,69,70,71,72]. A recurring objective in the reviewed research is to explore how farmers understand and rationalize the concept of welfare in their herds and what factors influence these perceptions [73,74]. This line of research is crucial for developing policies and programs that are well-received and implemented by producers.

Some studies investigate specific programs, such as the “Initiative Animal Welfare” (IAW) in Germany, which aims to establish higher standards of animal welfare [27,75]. These studies seek to understand why certain programs resonate positively among breeders, which can guide the creation of similar initiatives in other regions.

The assessment of attitudes among professionals in the swine and poultry industry regarding animal welfare in different farming systems is also a common theme [12,76]. This reflects the need to understand how management practices and the perceptions of professionals directly impact animal welfare, where the direct and indirect experience of swine producers also plays a crucial role in decision-making [77].

The Human–Animal Relationship (HAR) in different swine farming contexts is another important focus, with research evaluating how producers’ attitudes influence animal behavior, their reactions to humans, and how this relates to welfare and productivity [73,78]. The swine production chain is increasingly pressured to adopt regulations aimed at animal welfare, driven by ethical factors and the demands of both domestic and inter-national markets [75].

Research also focuses on public opinion, investigating how citizens evaluate different swine housing systems and their willingness to compromise animal welfare in trade-off situations [69,79,80]. The societal perception of swine production and how PLF influences the market is an emerging trend. With increasing automation and the use of advanced technologies in livestock farming, it is important to understand how these changes are perceived by consumers and the impact on their consumption choices [81,82].

Research on animal welfare certifications and the different requirements imposed by them is also a growing area. Studies seek to identify consistencies and differences in the welfare standards required, which can help harmonize and improve these certifications in a global context [76].

The studies reveal a growing concern for animal welfare in swine production, both from the perspective of producers and society. The identified trends indicate a movement towards adopting more ethical and sustainable practices, supported by specific programs, certifications, and PLF technologies. These studies are fundamental for advancing swine production towards a more conscious model, aligned with consumer and societal expectations [12,76,81].

Studies evaluate metrics compatible with Life Cycle Analysis (LCA) to assess agricultural systems [70,71,72,73,74,75,76,77,78,79,80,81,82]. LCA is a methodology that assesses all stages of a product, providing a comprehensive view of environmental, economic, and social aspects throughout the entire cycle [83]. In Precision Livestock Farming (PLF), LCA can be used to evaluate and optimize each stage of the animal production process, from animal feeding and management to the processing and distribution of products. However, assessing animal welfare still faces challenges, such as the standardization of metrics and the acceptance of proposed indicators by producers. There is a growing trend toward the adoption of harmonized terminology and labeling, aiming to make communication more transparent and reliable [70,73]. This is seen as an important step to ensure that products are perceived as “animal-friendly” and meet animal welfare expectations [3,73].

Government policies vary significantly between countries, influencing animal welfare practices [72,74,79,84]. The future of swine production seems to point towards greater integration of advanced technologies, combined with the need to develop reliable and applicable metrics in commercial environments.

### 4.3. Thematic Group C

Thematic group C analyzes “Animal Welfare,” particularly in relation to its understanding within PLF, revealing an increasing focus on several key areas that reflect emerging trends in the sector [20]. These studies provide a solid foundation for developing more ethical and sustainable practices in animal production while emphasizing the need to balance technological innovation with ethical concerns regarding animal welfare [1,85,86].

There is growing concern about environmental quality in swine facilities, reflected in various studies focused on evaluating environmental conditions and their impact on animal health [11,85,87]. These studies aim not only to review and improve air quality and ventilation but also to adapt management practices to established guidelines, such as those from Classy Farm, ensuring that the environment is conducive to pig welfare [85]. This focus on improving the environment highlights the importance of healthy living conditions as the foundation for responsible management.

Another highlight is behavioral monitoring of pigs, which increasingly benefits from the use of advanced technologies such as accelerometers and automated monitoring systems [62,65,86,87,88,89]. These devices allow for precise and continuous assessment of animal welfare, identifying behavioral changes that may indicate discomfort or stress [90,91,92,93]. The reliability of these tools is constantly evaluated to ensure that the collected data are consistent and relevant for decision-making in farm management [85,86,94].

Studies on animal behavior, such as the use of the Theoretical Domains Framework (TDF), provide insights into barriers to effective welfare management, such as tail biting [85,86]. The TDF helps identify effective behavioral interventions, suggesting that changing practices can lead to significant improvements in animal welfare.

The application of deep learning algorithms, such as Convolutional Neural Networks (CNN) and Kalman filters, is being explored to monitor and track animal movements, such as detecting posture and social behavior [4,88,92,95,96]. These systems are being im-proved to ensure that the analysis is accurate, even in large-scale production environments.

Despite advances, existing monitoring systems do not yet offer a complete multidimensional integration for a holistic assessment of animal welfare [93]. Studies point to the need for developing technologies that combine different data sources to provide a com-prehensive view of the animals’ condition, overcoming the limitations of systems that rely only on simple tracking algorithms [4,20,89,94].

Some farmers perceive that welfare practices may reduce production efficiency due to costs and additional work [20]. However, studies are challenging this view, demonstrating that the implementation of precision animal welfare technologies can, in fact, in-crease sustainability and production efficiency by minimizing medication use and optimizing environmental conditions for animals [11,85,94]. However, some studies also highlight the potential risks, weaknesses, and threats of this technology, suggesting careful adoption that considers the impacts on animal welfare and the sustainability of the implemented practices [20,65,89,94,97].

Furthermore, the welfare of pigs in slaughterhouses is another recurring theme, with studies comparing welfare indicators at different stages of the animal’s life, from gestation to slaughter [97,98,99,100]. This line of research aims to ensure that welfare standards are maintained throughout the animal’s life, promoting more humane treatment.

The reviewed studies indicate an evolution in PLF, with a growing focus on practices that balance technological innovation and ethics, always aiming for pig welfare. The incorporation of these technologies promises to transform the industry, making it more sustainable and responsible while ensuring better living conditions for animals.

### 4.4. Thematic Group D

Thematic group D, which analyzes “Productive and Economic Management” in the context of PLF, reflects a strong emphasis on using advanced technologies and data analysis to optimize productivity and the economic sustainability of swine production systems [101,102,103]. The incorporation of AI, automated systems, and adapted metrics reflects a growing commitment to innovation, animal welfare, and environmental responsibility—factors that will be decisive for the success and viability of swine farming in the coming years [101,102,103,104,105,106].

In the first place, there is a strong integration of new technologies, such as AI and automated monitoring systems, which are being applied to predict risks and improve pro-duction efficiency [102,105,106,107]. These robust systems are designed to operate in challenging environments and adapt to future needs, ensuring continuous and efficient production [101,102,103]. The use of AI, in particular, has proven effective in the early identification of risks to animal welfare, allowing proactive intervention before major problems arise [101,102,107,108].

Moreover, Industry 4.0, or the so-called Agriculture 4.0, is playing a crucial role in the transformation of livestock farming [105]. The potential integration of digital and automated systems can lead to a true revolution in the field, enabling smarter and more connected agricultural operations. This not only improves productivity but also promotes more sustainable resource management, reducing the environmental impact of operations [101,102,103,104,109].

One of the primary focuses of these studies is the development of metrics that integrate animal welfare with Life Cycle Assessment (LCA). By applying these metrics to various production systems, researchers aim to understand how different practices impact both productivity and sustainability [109]. This method, widely used to assess the environmental impact of products, is being adapted to include animal welfare as a crucial criterion, enabling a more holistic analysis of production systems [103,109].

Sustainability is also a central issue, with studies focusing on identifying bottlenecks and implementing solutions that can increase the sustainability of swine production systems [102]. The development of low-cost technologies that consider both animal welfare and the quality of life of producers is an important trend to make livestock farming more accessible and sustainable in the long term [103,105].

Regarding animal welfare, there is a growing effort to demystify the perception that promoting welfare reduces production efficiency. The development of Precision Animal Welfare (PAW) technologies challenges this view, showing that it is possible to reconcile both aspects [103]. Technologies such as camera-based monitoring and deep learning algorithms have proven promising, providing economical solutions to monitor pig behavior and improve management without compromising productivity [101,104,106,109,110].

Additionally, the issue of antimicrobial application has been addressed with a focus on improving management and prevention as key strategies for reducing its use [107]. The suggested approach is the personalization of management actions based on the lowest indices of each process, reinforcing the importance of clear instructions and specific training for farmers [102,107].

Another crucial aspect is the impact of pollutants in swine production. Studies indicate that the interaction of factors such as dust and ammonia can cause significant harm to animals, underscoring the need for rigorous environmental management to mitigate these harmful effects. This emphasizes the importance of sustainable practices that minimize pigs’ exposure to these pollutants [101,105].

Finally, the use of technologies such as deep learning algorithms for monitoring pigs has emerged as a trend, facilitating the characterization of animal behavior in an accessible and efficient manner [103,106]. This approach not only improves herd management but also contributes to the collection of valuable data that can be used to enhance animal welfare and productivity.

### 4.5. SWOT Analysis

The SWOT matrix was created based on an analysis using criteria that identify in-ternal and external factors affecting PLF in swine production [46,47]. The strengths reflect internal positive aspects and advantages that precision swine farming offers, while the weaknesses represent internal limitations (Table 6). Opportunities are external influences that can be exploited for future benefits, and threats are external factors that can harm the success of precision swine farming (Table 7). This analysis highlights the main internal and external factors influencing precision swine farming, providing a clear view of the strengths, weaknesses, opportunities, and threats, which can help in strategic decision-making to maximize benefits and mitigate risks.

PLF in swine production has several strengths, such as the adoption of advanced technologies (IoT, neural networks, cameras), which improve monitoring and efficiency, promote ethical and sustainable practices, and enhance both productivity and animal welfare. However, it faces weaknesses such as the lack of robust metrics to validate animal welfare, producer resistance due to costs and complexity, and the dependence on advanced technological infrastructure without global standardization. The opportunities include the advancement of autonomous and AI technologies, the growing demand for welfare certifications, and the potential to reduce antimicrobial use. On the other hand, threats such as inadequate technology implementation, variations in government policies, and environmental pollution pose challenges to the success of PLF.

In the reviewed studies, no management system incorporating PLF technologies covered the entire production process. Instead, these systems focus on specific processes, suggesting a significant gap for future academic developments. This direction points to the need for research that integrates the concepts of horizontal and vertical integration of Industry 4.0, enabling more efficient management and decision-making based on data and market trends [78,111,112]. Studies highlight that the application of technologies such as IoT, and AI has the potential to transform swine production into a more efficient and sustainable model, with decisions based on real-time data [16,107].

The enhancement of commercial systems to facilitate the implementation of PLF in the productive sector is highlighted in several studies. The collaboration between the equipment industry, management systems, and the academic sector offers a vast field for applied research [6,88,111,112,113,114,115]. The integration of these sectors promotes a continuous cycle of innovation, where academia can provide analyses for the development of technologies that meet market needs [101]. However, none of the reviewed studies mentioned the generation of product patents, raising questions about the goals and direction pursued by academia in developing new technologies.

This lack of patent records may indicate a greater focus on exploratory studies or practical applications rather than on the commercialization of technological innovations, which could limit the impact of these technologies on the industrial and agricultural sectors, and the ability to meet farmers’ needs positively [40,111]. It is essential that future research balances theoretical development with practical application to ensure that academic innovations result in concrete advances for the swine industry.

## 5. Conclusions

Precision Livestock Farming (PLF) stands out as an essential approach to addressing the current challenges in swine production, particularly regarding animal welfare, production efficiency, and environmental sustainability. The inclusion of technologies that are part of conventional technical development, such as environmental monitoring systems and ventilation devices, gains renewed relevance within the context of PLF by enhancing real-time monitoring and the precise management of animal welfare.

However, implementing these technologies faces considerable challenges, including producer resistance due to high costs and operational complexity. This barrier, coupled with the need for validation of welfare indicators and the lack of global standardization, limits the full and effective adoption of PLF. The absence of patent records also suggests that academia should aim to transform these innovations into more accessible commercial solutions for the sector.

The SWOT analysis conducted highlights both the advantages and limitations of PLF, emphasizing that, for this approach to succeed, it is crucial to balance theoretical development with practical application. Thus, it is expected that technological innovations will truly meet market and consumer expectations, promoting a more efficient, ethical, and sustainable swine industry aligned with global demands for responsible farming practices.

## Figures and Tables

**Figure 1 animals-15-02138-f001:**
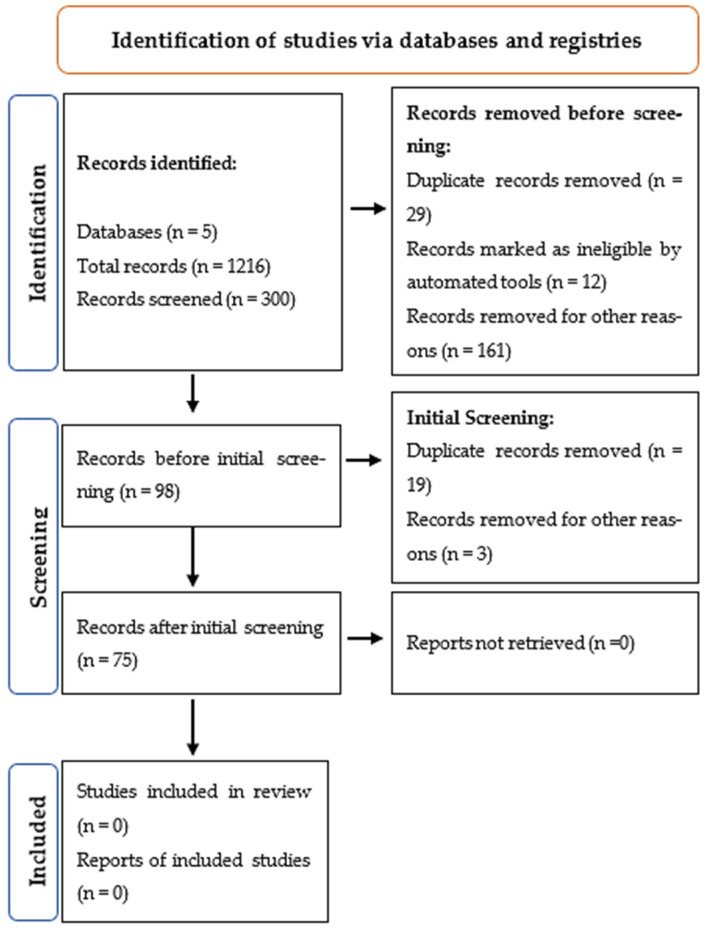
Modified PRISMA flow diagram of the systematic review (source: based on [49] Moher et al., 2009).

**Figure 2 animals-15-02138-f002:**
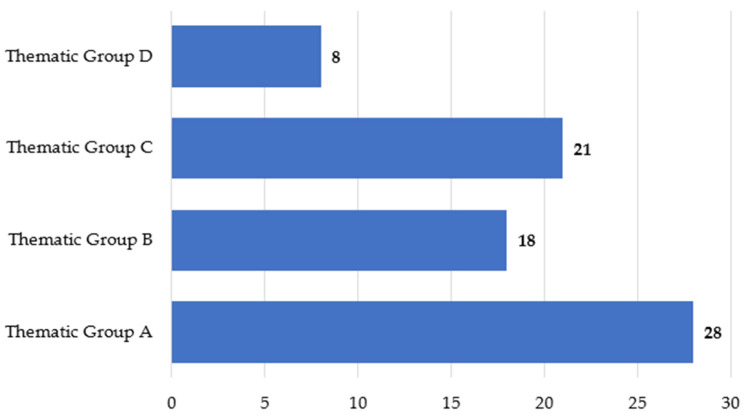
Number of articles researched by thematic group.

**Figure 3 animals-15-02138-f003:**
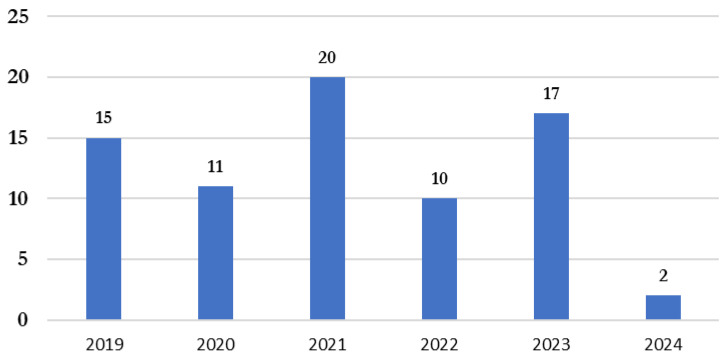
Temporal analysis of publication distribution (% of researched articles).

**Table 1 animals-15-02138-t001:** Thematic groups defined for the classification of articles.

Thematic Group	Main Approach	Subdivisions
A	Animal identification and monitoring	Automated identification technologies, individual animal tracking, monitoring, feeding behavior, and health
B	Human–animal relationships in swine farming	Understanding and use of concepts and techniques by producers and society, certifications for animal welfare
C	Animal welfare	Environmental conditions in facilities, thermal comfort, ventilation, floor layout, thermal stress, respiratory and heart rate monitoring, animal position, natural behaviors (rooting, locomotion, social interaction), and best practices
D	Productive and economic management	Economic data analysis and productivity indicators, financial break-even analysis for each animal, Industry 4.0 concepts and management systems applied to swine farming using PLF

**Table 2 animals-15-02138-t002:** Areas of focus addressed in researched articles.

Area of Focus	Number of Studies
Technologies for PLF	25
Application of Research Forms	15
Survey of the Producer or Consumer Perspective on PLF	14
Monitoring Systems	10
Management	10
Behavioral Aspects	7
Information Systems and Artificial Intelligence (AI)	3

**Table 3 animals-15-02138-t003:** Technologies addressed in researched articles.

Technology Addressed	Number of Studies
Image Analysis	10
Neural Networks and AI	5
Algorithms	5
RFID, IoT, and Sensors	5
Vocalization	3
Management and Monitoring	3

**Table 4 animals-15-02138-t004:** Methodologies addressed in researched articles.

Methodology Addressed	Number of Studies
Latest Advances	37
Development and Comparison of Models	26
Controlled Precision Observational Study	6
Research and Data Analysis	6

**Table 5 animals-15-02138-t005:** Types of results achieved in researched articles.

Type of Result	Number of Studies
Indicators of Animal Welfare in Pigs	30
Equipment and Systems Qualification	20
Social Acceptance of PLF	11
Pig Behaviors	10
PLF Systems Reliability	4

**Table 6 animals-15-02138-t006:** SWOT matrix of PLF in swine production—strengths and weaknesses.

Strengths	Weaknesses
Integration of advanced technologies such as IoT, neural networks, and RGB cameras that increase precision in animal monitoring and identification [50,52].	Challenges in validating positive animal welfare indicators, such as affective states, limiting the full effectiveness of the technologies [57,67].
Significant contribution to more ethical and sustainable practices, aligned with consumer and societal expectations [20,73].	Resistance from producers to adopt technologies due to high initial costs and perceived operational complexity [20,64].
Increased production efficiency through real-time monitoring and automation of processes [105,107].	Dependence on large volumes of data and the need for advanced technological infrastructure, which may be inaccessible to small producers [101].
Improved animal welfare through systems that detect ab-normal behaviors and track health status [59,93].	Lack of standardized metrics for assessing animal welfare and limited acceptance by producers [76].

**Table 7 animals-15-02138-t007:** SWOT matrix of PLF in swine production—opportunities and threats.

Opportunities	Threats
Expansion of autonomous technologies and AI systems to promote more sustainable and efficient swine farming [107,109].	Risks associated with the lack of validation of animal welfare indicators and inadequate implementation of technologies, affecting effectiveness and acceptance [91].
Greater demand for animal welfare certifications and sustainable practices in both domestic and international markets [73,81].	Regulatory pressure and government policies can vary significantly between regions, negatively influencing the adoption of precision technologies [72].
Potential to reduce the use of antimicrobials and improve animal health through more efficient and proactive management [107].	Pollution, such as dust and ammonia, directly affects the health of pigs, representing a significant environmental challenge [101,105].

## Data Availability

The datasets generated and/or analyzed during the current study are available upon request to the corresponding author.

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
