# Peer review of "Precision Livestock Farming Applied to Swine Farms—A Systematic Literature Review"

_animals, 2025, doi:10.3390/ani15142138_

Round 1

Reviewer 1 Report

Comments and Suggestions for Authors

1. The presented article is focused and progressive methods referred to as Precision Livestock Farming (PLF), applied to pig farming. The article is processed systematically, but I have several comments and questions about it.

2. Keywords need to be modified according to the guidelines for authors.

3. In the first part of the Introduction, the authors gradually deal with the most significant methods, which are a significant contribution in terms of improving animal welfare, and which bring economic benefits to farmers. They base their brief comments on 28 scientific publications.

4. The second part of Materials and Methods is devoted to considerations and a description of the methodology of creating a Review article. The authors base this description on 2 pages of text from other publications (29 to 49).

5. On line 138, the title of Table 1 is in Portuguese, it should have been translated into English.

6. The third chapter of Results deals with the systematic classification of articles and their division into groups according to content focus.

7. The article contains a large number of typos and deficiencies of a formal nature, for example in Tables 3, 4, 5. The words in these tables are separated by a separator, a hyphen.

8. In terms of the focus of the PLF article, the structure proposed in the paragraph on lines 180 to 183 is a bit unclear. "The data were categorized into eight distinct areas: Technologies for Precision Livestock Farming, Forms, Producer or Consumer Perspective, Monitoring, Management, Behavioral, Information Systems, and AI. See works Table 2. It is not entirely clear why the term "Forms" was included in this structure? Please explain what it brings new and interesting? Similarly, the inclusion of Forms in the structure and Table 3 is unclear.

9. The Discussion chapter provides an overview of the use of PLF in pig farming from the point of view of various areas, mentioned in the previous chapters and divided into several groups. Considerable attention is paid to the issue of animal welfare.

10. The conclusion summarizes the content of the previous chapters. Some of the evaluated research activities and technical equipment are part of the normal technical development of buildings and their equipment. The question is whether the inclusion of these areas in the PLF is appropriate. The authors rightly point out the problems and limitations arising from the economic and social conditions in the given areas.

Reviewer 2 Report

Comments and Suggestions for Authors

I found an inconsistency in the abstract: the use of the term "investment in research and development." In one instance, it is referred to as "the need for increased research and development investment," while in another, a similar phrase is used with slight variation, which can lead to redundancy and confusion. Standardizing this term to "increased investment in research and development" would enhance clarity and readability.

Adjusted the structure of the article to make it more fluid, ensuring it reads more like a narrative with "however, challenges remain" linking benefits with areas needing improvement.

Introduction: 

The purpose and scope are somewhat repetitive, such as restating the goal of the literature review and the focus on animal welfare several times. Streamlining these sentences can clarify the focus without redundancy.

The phrase “state of the art” is used without context, which might be unclear to some readers. Rephrasing to “current state of research” or “latest advancements” could improve comprehension.

The introduction could benefit from a clear outline of the literature gaps earlier. Stating the challenges (like lack of standardized welfare metrics and acceptance by farmers) more prominently at the beginning can strengthen the argument for the study’s necessity.

Methodology:

The methodology uses "PLF" (Precision Livestock Farming) inconsistently. Standardizing terms, such as using "PLF" throughout once introduced, would improve clarity.

Some phrases, such as "the search terms included," could benefit from added clarity about why specific terms were chosen, enhancing methodological transparency. Explaining the rationale for selected databases and specific inclusion/exclusion criteria would strengthen the section.

The text mentions both "qualitative coding methods" and "meta-analysis" without clearly defining how these analyses integrate. Providing more detail on these integration methods would help readers better understand the mixed-methods approach.

Results:

Descriptions of the categories and themes could benefit from clearer phrasing. For example, phrases like "Management and monitoring is the least addressed area" could be rephrased for better flow: "Management and monitoring were the least researched areas."

Terms like "IoT" (Internet of Things) and "RFID" (Radio-Frequency Identification) appear without prior definition. Adding these definitions at first mention would help ensure understanding for readers unfamiliar with the terms.

The statement "Management and monitoring is the least addressed area" could be expanded to provide specific percentages or study counts for better quantitative context.

Conclusions:

The term "Precision Livestock Farming" should be consistently abbreviated as "PLF" after its initial introduction, but there are places where the full term is repeated unnecessarily.

The section restates challenges like "lack of standardized metrics" and "producer resistance" multiple times. Summarizing these challenges more succinctly would streamline the conclusion.

The conclusion references strengths, weaknesses, opportunities, and threats without explicitly linking these to the specific findings in the SWOT analysis. Clarifying each SWOT component of the study’s results would provide a clearer takeaway for readers.

Reviewer 3 Report

Comments and Suggestions for Authors

Your paper is interesting and deals with innovative topics. I have attached a file with some suggestions that could improve the framework. The main problem, in my opinion, is an approach to sustainability that needs to be deepened and clarified. After overcoming this (quite important) weakness, the manuscript is more than satisfactory.

Regards

Author Response

The responses to the reviewer's comments are in the attached file.

Round 2

Reviewer 2 Report

Comments and Suggestions for Authors

In my opinion, the manuscript can be published.

Author Response

We appreciate the contributions made to improve our article.

Reviewer 3 Report

Comments and Suggestions for Authors

Dear authors,

The improvement of your paper is satisfactory

Regards